# Adherence to Covid-19 preventive measures among high school students in Jimma town, South-West Ethiopia: Institutional-based cross-sectional study

**Belete Fenta Kebede**[1]*, **Yalemtsehay Dagnaw Genie**[2], **Tsegaw Biyazin Tesfa**[1], **Aynalem Yetwale Hiwot**[1], **Asiya Muhhamedamin Abagelan**[1], **Mulualem Silesh Zerihun**[3]

**1** School of Midwifery, Jimma University, Jimma, Ethiopia, **2** Department of Nursing, Mizan-Tepi University, Mizan-Aman, Ethiopia, **3** Department of Midwifery, Debre-Berhan University, Debre Berhan, Ethiopia

☯ These authors contributed equally to this work.
* belete121wy@gmail.com

**Data Availability Statement:** All relevant data are within the paper and its Supporting Information files.

## Abstract

### Background

Corona (COVID-19) is an infectious disease caused by a newly discovered corona virus. The World Health Organization has recommended several preventive measures for COVID-19 and African countries, including Ethiopia had accepted and engaged in the recommended preventive measures. Adherence to COVID-19 prevention measure is still a big problem; however, the level of adherence to preventive measures had not reported in Ethiopia among students and there is an information gap, therefore, this study conducted to fill the information gap on level of adherence to COVID-19 preventive measures among students.

### Objective

This study aimed to assess the level of adherence to COVID-19 preventive measures and its associated factors among high school students in Jimma Town public High Schools in southwest Ethiopia, 2021.

### Methods sand materials

An institution-based cross-sectional study was conducted among 404 systematically selected high school students from Jimma town from July 15 to August 2, 2021. The sample size was determined using a single-population proportion formula, and data were collected through face-to-face interviews using a structured and pretested questionnaire. Data were entered into Epi-data manager 4.4.2.1 then exported to Stata 14 for cleaning and analysis. Bivariate and multivariable ordinal logistic regression analyses were declared to identify significant variables. Finally; significant factors were determined at a significance level of <0.05.

**Funding:** The author(s) received no specific funding for this work.

**Competing interests:** The authors have declared that no competing interests exist.

**Abbreviations:** COVID-19, corona virus disease 2019; AOR, adjusted odds ratio; COR, crude odds ratio; WHO, World Health Organization; PPE, personal protective equipment; SARS, severe acute respiratory syndrome; IPC, infection prevention and control.

## Results

Of 388 students included in the analysis, approximately 14.7% (95%CI: 11.51–18.60) of students had good level of adherence to COVID-19 preventive measures.Only 6.9%of participants had good knowledge where as approximately half of the respondents had favorable attitude toward COVID-19 preventive measures. Factors such as Female gender (AOR = 1.03(95%CI: 1.01–1.74), access to water and soap (AOR = 2.11(95%CI: 1.06–4.19) andattitude (AOR = 4.36(95%CI: 2.69–7.08)) were found to have a statistically significant association with level of adherence to COVID-19 preventive measures.

## Conclusion

Adherence to COVID-19preventive measures among students wasunexpectedly lower than in other studies. Female gender, lack of access to water and soap, and attitudes were factors associated with adherence to COVID-19 preventive measures. Therefore, to ensure maximal adherence to preventive measures for COVID-19, special messages and efforts targeting males, increasing access to water and soap, trainingto improve attitude toward COVID-19 preventive measures should be implemented at schools.

## Introduction

The global outbreak of the COVID-19 pandemic has spread worldwide, affecting almost all countries and territories. Corona virus disease 2019 (COVID-19) is a respiratory illness that affects the human respiratory system [1].

The World Health Organization (WHO) has recommended several preventive measures for COVID-19, such as regular hand washing with water and soap, social distancing, wearing masks, covering the mouth while coughing and sneezing, and avoiding touching the eyes, nose, and mouth. African countries including Ethiopia have accepted and engaged in the recommended preventive measures [2–4]. The impact of the pandemic, loss of livelihoods and school closures have far-reaching effects on children's and young people's health, safety, and education [5]. Up to 99% of students were in low-and lower-middle-income countries, and Some 23.8 million additional children and youth (from pre-primary to tertiary) may drop out or not have access to school next year due to the pandemic's economic impact alone [6].

By the end of 2020, school closures affected approximately 139,000,000 learners [7].The COVID-19 pandemic affecting nearly 1.6 billion learners in more than 200 countries and the suspension of face-to-face instruction in schools,which hasconsequences for students' learning [8]. By the end of May 2020, 20 school systems had partially opened, and approximately 1.2 billion students remained out of school [9].

Immediately after the first confirmed case of COVID-19 in Ethiopia on 13 March 2020, three days later, on 16 March 2020, the Office of the Prime Minister of Ethiopia announced that schools, sporting events, and public gatherings would be suspended for 15 days. This was followed by the declaration of a state of emergency that lasted for five months (April–August 2020) [10, 11].

Around November 2020, the Ethiopian government announced the reopening of schools to grades8 and 12 students, who needed to sit for national examinations with safety guidelines [12]. Gradually, all schools had started face education and students were provided with a

supply of facemasks, but the availability of sanitation facilities was limited to schools in Ethiopia [13].

To the best of our knowledge, few studies have been conducted on the level of adherence and factors affecting students' adherence to COVID-19 preventive measures. Therefore, this study aimed to assess adherence to COVID-19 preventive measures and associated factors among high school students in Jimma town, southwest Ethiopia.

## Methods and materials

### Study area and period

The study was conducted in High schools in Jimma Town, Oromia regional state of Ethiopia. Jimma Town is located 355 km away from Addis Ababa in the southwest direction. The Town has six government High Schools;Jiren High School, Seto High School, Kera High School, Aba Buna High School, Geda High School, and Jimma town secondary. Of these, threewere selectedusing the lottery method (Jiren High School, Geda High School, and Jimma town secondary). Jiren High school serves 1828 students, Geda High School) serves 1475 students and Jimma Town Secondary School serves 999 students. The study was conducted from July 15to August 2, 2021.

### Study design

This institution-based cross-sectional study was conducted among High School students in Jimma Town public High schools.

### Population, eligibility criteria

All High School students following their education in Jimma Town High Schools in 2021were the source population. All selected High School students (Grade 9th and 10th) in Jimma Town High Schools during data collection were study population and they were eligible for the study.

### Sample size determination

The sample size was determined using single population proportion formula by considering the following statistical assumptions: from previous study conducted in Gondar, proportion (P) = 51.03% [2], 95% confidence interval (Z $\alpha$/2 = 1.96), $\alpha$ = 0.05 and 5% marginal of error.

n = (z @/2)2*p(1−p)

n = (1.96)2 *0.51(1−0.51) / (0.05)2

n = 0.954 /0.0025 = 384. After adding a nonresponse rate of 5% (19.2~20) = 404.

### Sampling techniques and procedure

From the six public high schools found in Jimma Town, three High Schools (Jiren, Geda, and Jimma town secondary were selected using the lottery method. Students were selected using the admission registration number from their roster and samples were proportionally selected from each high school. A systematic sampling technique was used to select study participants from three sampled high schools, with a total population of 4302. The interval (K$^{th}$) was determined by dividing the total population (4302) by the final sample size (404),which was approximately 11. The first study participant was selected by the lottery method using their registration serial number, and the rest were selected at intervals of 11 students, until the final sample size was reached.

## Data collection instrument and quality control

After reviewing previous studies [1], [2], [14–16],a questionnaire was adapted to address the objectives of the study. To ensure the quality of the data, the tool was first prepared in English and then translated into the local languages (Amharic and Afan Oromo) by language experts in English and both local languages. The reliability and validity of the instrument was assessed using Cronbach alpha (0.751) and subject expertise (public health $ nursing) respectively. A pre-test done on 5% of the total sample size in another school, which was not selected for actual data collection. Modifications such as wording, rephrasing, adding and deleting some information for clarity were made to the tool accordingly. Data collectors and supervisors were trained in the data collection process for one day. Data were checked for completeness and consistency of the information provided by the principal investigator.

## Variables of the study

The dependent variable was adherence to COVID-19 preventive measures.

The independent variables included sociodemographic variables (sex, age, grade level, residency, ethnicity and religion). Respondents' knowledge, attitude, practice of COVID-19, COVID-19 exposure status of students, and facility-related variables (availability of water, mask supply, sanitizer, and training).COVID-19 preventive measures include hand washing, use of face cover/face masks, maintaining physical distancing, avoiding touching the face, and cleaning or disinfecting frequently used surfaces.

## Operational definitions

**Knowledge.** Refers to the concept of students regarding COVID-19 and understanding COVID-19 preventive measures. There were 14(fourteen) knowledge measuring questions, A score of <50%was considered as they have moderate knowledge 59–79% and respondents scored 80–100%, were considered as they have good knowledge [17].

**Attitudes.** Refers to what the respondent thinks of correctly regarding COVID-19 prevention measures in the questionnaire prepared by the researcher. To assess attitude,5(five) questions with Likert scale items of (1) strongly disagree, (2) disagree, (3) neutral, (4) agree and (5) strongly agree were used. Number of responses was calculated and the score was considered who have favorable attitude if they had score ≥50%of the total score and unfavorable attitude if they had score <50%of the total score on scale [2, 17].

**Adherence.** Refers to the concept of students regarding COVID-19 and compliance with COVID-19 prevention measures and which were generated from hand washing, using a facemask, maintaining physical distance, not traveling to a crowded place, home stay, and not traveling to anyplace outside the city in the last two weeks. Six (6) items questions were used to assess the adherence level. Then, respondents who scored less <50% measured as to have a poor level of adherence, Respondents who scored 50–79% were considered as have moderate level of adherence and respondents who scored 80%-100% were considered as they have a good level of adherence to COVID-19 preventive measures [2, 17, 18].

## Data processing and analysis

After checking data completeness and consistency, the collected data were coded and entered into Epi-data version 4.4.2.1(www.epidata.dk/download.php) and exported into Stata version 14 (www.stata.com) for cleaning and analysis. Descriptive statistics were used to generate frequency tables and graphs and ordinary logistic regression modeling was used to estimate crude (p<0.25) and adjusted odds ratios (p<0.05). During the analysis, we used the overall

level of adherence as the dependent /outcome. The independent variables with a p-value less than 0.05 were considered statistically significant with the outcome variable. Data were collected anonymously to maintain the confidentiality of the study participants.

## Ethics approval and consent to participate

An ethical clearance letter was obtained from the School of Midwifery, Institute of Health, College of Health Sciences Jimma University. Permission letters were written for three high schools (Geda, Jiren, and Jimma High Schools). Permission was obtained from high schools. Data were collected after obtaining written consent from the high school directors after explaining the purpose of the study. Oral informed consent was obtained from all study participants (students) aged greater than 18 years for ethical acceptability and consent was obtained from the parents or guardians of the minors / those aged <18 years included in the study. Finally, they voluntarily participated in the interviews and had the right to withdraw at any time, without any penalties. All data collected from the students were kept strictly confidential and were used only for study purposes. Students' names were not included inthe data-collection tool.

## Result

In this study from a total sample of 404 eligible participants, only 388 respondents were participated and provided complete information making a non-response rate of 96.03%. Six students refused to consent to participate in the study and ten students gave incomplete information.

## Socio-demographic characteristics of students

In this study, among the total respondents of 388, 188(48.5%) of students were male and 200 (51.5%) were female. Two hundred nineteen (56.4%) of participated students were Muslim, 96 (24.7%) followed by Orthodox, 68(17.5%) religious followers. The majorities (78.4%) of the students in this study were from urban and only (21.6%) were from rural residences (Table 1).

## Students' level of knowledge and attitude towards COVID-19

In this study, the majority of students had poor knowledge of covid-19(70.4%), but approximately 67.5% of the respondents had a favorable attitude towards covid-9 prevention measures (Table 2).

## Preventive practices of high school students towards covid-19

This study indicates that most (88.4%) of high school students had washed /keeping hand hygiene using water, hand rubs, and sanitizers. More than half (52.1%) of the students maintain their social distancing and the majority (62.6%) of students used facemasks /covered their faces (Table 3).

## Institutional factors of COVID-19 preventive practices

In this study majority, 78.6%) of the respondents indicated that the presence of hand washing practices. However, 83(21.4%) of the responses of the respondents indicated that the absence of a hand washing facility and 58% of the respondents revealed the presence of water and soap in their school. However, a significant number of respondents 163 (42%) reported that there was an absence of water and soap.

Approximately 225 (58%) of the respondents revealed that the arrangement of the chair was by keeping the recommended distance of 2meter. On the other hand, 163(42%) of the respondents justified the absence of arrangement of tables and chairs by keeping the recommended distance of 2m (Table 4).

**Table 1. Socio-demographic characteristics of students of Jimma high school, Jimma town southwest Ethiopia, 2021 (N = 388).**

| Characteristics | Category | Frequency | Percentage |
|---|---|---|---|
| Sex | Male | 188 | 48.5 |
| | Female | 200 | 51.5 |
| Age | <18 | 359 | 92.5 |
| | ≥18 | 29 | 7.5 |
| Ethnicity | Amhara | 95 | 24.5 |
| | Gurage | 43 | 11.1 |
| | Oromo | 204 | 52.6 |
| | Tigre | 28 | 7.22 |
| | Others* | 18 | 4.7 |
| Religion | Catholic | 5 | 1.3 |
| | Muslim | 219 | 56.4 |
| | Orthodox | 96 | 24.7 |
| | Protestant | 68 | 17.5 |
| Residency | Urban | 304 | 78.4 |
| | Rural | 84 | 21.6 |
| Educational status | Grade 9 | 197 | 50.8 |
| | Grade 10 | 191 | 49.2 |
| History of medical illness | Yes | 38 | 9.8 |
| | No | 350 | 90.2 |

*: Yem,Somali,Kefa and Silte.

## Adherence with COVID-19 preventive measures

In this study, the overall level of adherence towards COVID-19 preventive measures, approximately, 47.2% (95%CI: 42.2–52.26, 38.1% (95%CI: 33.4–43.1) and 14.7% (95%CI: 11.51–18.60), of students had poor, moderate and good level of adherence towards COVID-19 preventive measures, respectively (**Fig 1** and **Table 5**).

## Exposure status characteristics

In this study, only 63(16.2%) had a history been as casein COVID-19, and only 50(12.9%) students had their friends been a case of COVID-19 (Table 6).

## Factors associated with adherence towards COVID-19 prevention measures

The relationship between the dependent and independent variables were analyzed using an ordinary logistic regression model. After checking the assumption, collinearity, and fitness of

**Table 2. The overall level of knowledge and attitude towards COVID-19 among Jimma high school students 2021/ (n = 388).**

| Characteristics | Category | Frequency | Percentage |
|---|---|---|---|
| Knowledge of students | Poor Knowledge | 273 | 70.4 |
| | Moderate Knowledge | 88 | 22.7 |
| | Good Knowledge | 27 | 6.9 |
| Attitude of students | unfavorable attitude | 191 | 49.2 |
| | Favorable attitude | 197 | 50.8 |
| Total | 388 | | 100 |

**Table 3. The practice of COVID-19 preventive measures among students of Jimma high school, Jimma town south west Ethiopia, 2021 (N = 388).**

| Characteristics | Category | Frequency | Percentage |
|---|---|---|---|
| hand washing/keeping hand hygiene | No | 45 | 11.6 |
| | Yes | 343 | 88.4 |
| Maintaining social distancing | No | 186 | 47.9 |
| | Yes | 202 | 52.1 |
| wearing facemask/face covering | No | 145 | 37.4 |
| | Yes | 243 | 62.6 |
| Avoidance overcrowding conditions | No | 176 | 45.4 |
| | Yes | 212 | 54.6 |
| Home stay practice | No | 202 | 52.1 |
| | Yes | 187 | 48.9 |
| Traveling history to the crowded place over 2 weeks | No | 176 | 45.4 |
| | Yes | 212 | 54.6 |
| Total | | 388 | 100 |

the ordinary logistic regression model, variables with a significance level of<0.25 during bivariate analysis were transferred to adjusted multivariable ordinary logistic regression. Covariates such as sex, attitudes, knowledge of covid-19, institutional related factors such as presence of water and soap were significant at p-value< 0.25 in bivariate analysis and were eligible for adjusted multivariable regression analysis. Finally, from the adjusted multivariable analysis, covariates such as sex, presence of water and soap, and Attitudes towards COVID-19remained independent factors for adherence to COVID-19 preventive measures. In this study, The odds of having good adherence towards COVID-19 preventive measures were 1.03 times more likely among female high school students than among male high school students (AOR = 1.03 (95%CI: 1.01–1.74).

High school students who had a favorable attitude towards COVID-19 had 4.36 times higher odds of good adherence toCOVID-19 preventive measures than students who had unfavorable attitudes (AOR = 4.36 (95%CI: 2.69–7.08)).

The odds of adherence among students who had access to water and soap at school was 2.11 times more than among high school students who had no access to water and soap at school to adhere to COVID-19 preventive measures (AOR: 2.11(95%CI: 1.06–4.19) **(Table 7)**.

**Table 4. Institutional factors of COVID-19 preventive practice among Jimma high schools 2021 (n = 388).**

| Institutional characters | Category | Frequency | Percentage |
|---|---|---|---|
| Presence of hand washing facility | Yes | 305 | 78.6 |
| | No | 83 | 21.4 |
| Presence of water and soap | Yes | 225 | 58.0 |
| | No | 163 | 42.0 |
| Presence material promoting hand washing | Yes | 217 | 55.9 |
| | No | 171 | 44.1 |
| posters displayed many massages | Yes | 248 | 63.9 |
| | No | 140 | 36.1 |
| Arrangement of tables and chairs at least 2 meters | Yes | 225 | 58.0 |
| | No | 163 | 42.0 |
| Adequate ventilation | Yes | 243 | 62.6 |
| | No | 145 | 37.4 |
| Total | 388 | | 100 |

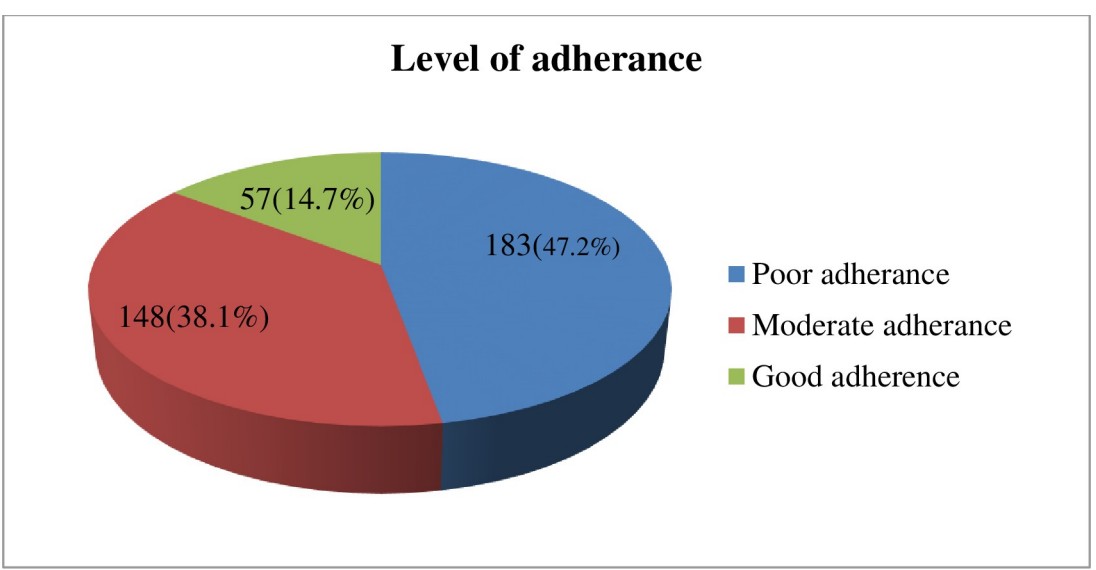

**Fig 1. Overall adherences, level of COVID-19 preventive adherences among students of Jimma public high schools /2021 (N = 388).**

## Discussion

This study determined the level of adherence and factors associated with COVID-19 preventive measures among high school students in southwest Ethiopia.

**Table 5. The level of COVID-19 preventive adherence students of Jimma town public high schools, southwest Ethiopia/2021 (n = 388).**

| Characteristics | Category | Level of adherence | | |
|---|---|---|---|---|
| | | Poor adherence N (%) | Moderate adherence N (%) | Good adherence N (%) |
| Sex | Male | 98(52.13) | 59(31.38) | 31(16.49) |
| | Female | 85(42.50) | 89(44.50) | 26(13.00) |
| Age of students | <18 | 173(48.19) | 132(36.77) | 54(15.04) |
| | ≥18 | 10(34.48) | 12(41.38) | 7(24.14) |
| Educational status | Grade 9 | 84(42.64) | 79(40.10) | 34(17.26) |
| | Grade 10 | 99(51.83) | 69(36.13) | 23(12.04) |
| Presence of hand washing facility | No | 49(59.04) | 25(30.12) | 9(10.84) |
| | Yes | 134(43.93) | 123 (40.33) | 48(15.74) |
| Presence of water and soap | No | 94(57.67) | 51(31.29) | 18(11.04) |
| | Yes | 89(39.56) | 97(43.11) | 39(17.33) |
| Presence of posters hygiene, and physical distancing | No | 90(52.63) | 63(36.84) | 18(10.53) |
| | Yes | 93(42.86) | 85(39.17) | 39(17.97) |
| Attitude of respondents | Unfavorable attitude | 129(67.54) | 55(28.80) | 7(3.66) |
| | Favorable attitude | 54(27.41) | 93(47.21) | 50(25.38) |
| Knowledge of respondents | Poor knowledge | 130(47.62) | 108(39.56) | 35(12.82) |
| | Moderate knowledge | 41(46.59) | 34(38.64) | 13(14.77) |
| | Good knowledge | 12(44.44) | 6(22.22) | 9(33.33) |
| A family member has been a case of COVID-19 | No | 156(47.27) | 127(38.48) | 47(14.24) |
| | Yes | 27(46.55) | 21(36.21) | 10(17.24) |
| Overall level of adherence n (%) | | 183(47.2) | 148 (38.1) | 57(14.7) |

**Table 6. COVID-19 exposures status among high school students of Jimma public high schools 2021 (n = 388).**

| Exposure status characteristics | Category | Frequency | Percentage |
|---|---|---|---|
| History of being a case of COVID-19 | Yes | 63 | 16.2 |
| | No | 325 | 83.8 |
| Any member of your family had been a case of COVID-19 | Yes | 58 | 14.9 |
| | No | 330 | 85.1 |
| Any of your friends been a case of COVID-19 | Yes | 50 | 12.9 |
| | No | 338 | 87.1 |

In this study, the overall rate of good adherence to COVID-19 preventive measures was approximately 14.7%. This is lower than other studies conducted in Gondar(51.04%) [2], Jeddah City(49%) [18], North Shoa Zone, Ethiopia 44.1% [19] and Hossana, South Ethiopia where only 50.4% of the study participants had good adherence to the COVID-19 preventive measures [20]. However this study finding is slightly higher than that of a study done in Dirashe District, Southern Ethiopia, where only 12.3% adhered to the recommended COVID-19 preventive measures [17] which may be due to differences in study settings and participants. The study setting in this study was high schools, which means that it was at the institutional level, whereas other studies were conducted at the community level. Our study involved only high school students where as other studies included all populations in the community.

In our study, students who had access to water and soap at school was 2.11 times more likely to adhere to COVID-19 preventive measures than high school students who had no access to water and soap at school (AOR = 2.11(95%CI: 1.06–4.19). This is comparable to a

**Table 7. Factors affecting COVID-19preventive adherence among students of Jimma town public high schools, southwest Ethiopia/2021 (n = 388).**

| variables | Category | Unadjusted and adjusted ordinary logistic regression | | | |
|---|---|---|---|---|---|
| | | COR | p-value | AOR | p-value |
| Age | <18 years old | 1.04(.28–3.92) | .953 | | |
| | >18 years old | 1 | | | |
| Sex | Female | 1.05(1.04–1.88) | .002 | 1.03(1.01–1.74) | .012* |
| | Male | 1 | | 1 | |
| Educational status | Grade 10 | 1 | | | |
| | Grade 9 | 1.74(.95–3.19) | .072 | | |
| Attitude | Unfavorable | 1 | | 1 | |
| | Favorable | 4.04(2.55–6.40) | .000 | 4.36(2.69–7.08) | .000* |
| Presence of water and soap | No | 1 | | 1 | |
| | Yes | 2.29(1.22–4.29) | .010 | 2.11(1.06–4.19) | .034* |
| Presence of hand washing facility | No | 1 | | | |
| | Yes | 1.95(.89–4.27) | .095 | | |
| Presence of posters hygiene, and physical distancing | No | 1 | | | |
| | Yes | 2.01(.12–3.93) | .021 | | |
| Knowledge of respondents | Good | 1 | | 1 | |
| | Moderate | .36(.14-.92) | .033 | .71(.25–2.03) | .521 |
| | Poor | .42(.15–1.23) | .113 | .74(.23–2.41) | .621 |
| Family history being case of COVID-19 | No | 1 | | | |
| | Yes | 1.23(.55–2.72) | .611 | | |

NB

* = significant at p value<0.05 in multivariable analysis, 1.00 = considered as reference categories = COR–crude odds ratio, AOR-adjusted odds ratios.

cross-sectional study conducted among health professionals in public hospitals in southeast Ethiopia [18]. Water is an essential tool for infection prevention and hand washing is a simple but important procedure for infection prevention.

This study found that the odds of having good adherence to COVID-19 preventive measures were 1.03 times more likely among female high school students than among male high school students (AOR = 1.03(95%CI:1.01–1.74). This finding was supported by studies conducted in Sudan [21] and Gondar [2]. A possible explanation might be that the majority of male students went outside their homes after school, and may have moved from one place to another (here and there).Therefore, preventive measures might not be kept as per standard at each place more often, and it may be difficult to comply with physical distancing. The other possible reason may be the habitual norms in Ethiopia, being women/females are allowed to do solely home activity and contrasts to this male delegate to outreach activity. Indirectly it has benefited through restricting and obeys the strategy of "stay at home", it is one method of prevention of COVID-19.

In our study finding, high school students who had favorable attitude towards COVID-19 had 4.36 times higher odds of good adherence to COVID-19 preventive measures than students who had unfavorable attitudes (AOR = 4.36 (95%CI: 2.69–7.08)). This study finding was supported by the study conducted in Northwest Ethiopia [22],Gonder [23],Dirashe District, Southern Ethiopia [20], Dire-dawa, Eastern Ethiopia [24],and systematic review and meta-analysis [15]. The possible explanation for this may be that a positive attitude should be a prerequisite for developing and practice new behavior.

## Limitation of the study

The following limitation can be considered in relation to this study. It is very clear that this is a cross-sectional study and the results could not infer causality. Social desirability bias may one limitation that students might respond to the socially acceptable responses despite their actual practice is poor. There is also a possibility of recall bias since we asked questions about having COVID-19 to themselves, their family, and their friends. In addition, it has been developed a cut-off value for attitude of respondents based on half score in each category, this can create some limitations to their measure.

## Conclusion

In this study, the over al level of good adherence to COVID-19 preventive measures was very low, which is only around seven percent. Female Sex, availability of water and soap at schools, and attitude towards COVID-19 prevention measures were independent predictors of students' adherence towards COVID-19 preventive measures. Different stalk holders (Ministry of Education, Ministry of Health. . .) should work to improve the level of adherence with COVID-19 preventive and control measures to stop its spread and minimize its disastrous impact. This study may draw the attention of policymakers/program managers to the urgent need for implementing a health education campaign targeting male students. It would also be worthwhile to invest on improving access to water and soap, and use of innovative strategies based on local evidences to raise the students' awareness and to improve their attitude to COVID-19 preventive measures.

## Supporting information

**S1 Checklist.**
(DOCX)

**S1 File. SPSS support data for PLOS.**
(SAV)

**S2 File. Questioner support data for PLOS.**
(DOCX)

## Acknowledgments

Jimma University College of Health Science, Institute of Health, School of Midwifery, acknowledged for providing this opportunity to conduct this study. Our thanks also go to Jiren High School, Geda High School, and Jimma Town secondary school directors/administrators and all Jimma High School students who participated in this study. It is also our pleasure to thank the data collectors and supervisors.

## Author Contributions

**Conceptualization:** Belete Fenta Kebede, Tsegaw Biyazin Tesfa, Aynalem Yetwale Hiwot, Asiya Muhhamedamin Abagelan, Mulualem Silesh Zerihun.

**Data curation:** Belete Fenta Kebede, Yalemtsehay Dagnaw Genie, Tsegaw Biyazin Tesfa, Aynalem Yetwale Hiwot, Mulualem Silesh Zerihun.

**Formal analysis:** Belete Fenta Kebede, Yalemtsehay Dagnaw Genie, Mulualem Silesh Zerihun.

**Investigation:** Yalemtsehay Dagnaw Genie, Tsegaw Biyazin Tesfa, Aynalem Yetwale Hiwot.

**Methodology:** Belete Fenta Kebede, Tsegaw Biyazin Tesfa, Aynalem Yetwale Hiwot, Asiya Muhhamedamin Abagelan.

**Project administration:** Tsegaw Biyazin Tesfa.

**Resources:** Yalemtsehay Dagnaw Genie, Mulualem Silesh Zerihun.

**Software:** Tsegaw Biyazin Tesfa, Mulualem Silesh Zerihun.

**Supervision:** Belete Fenta Kebede, Aynalem Yetwale Hiwot, Asiya Muhhamedamin Abagelan, Mulualem Silesh Zerihun.

**Validation:** Belete Fenta Kebede, Yalemtsehay Dagnaw Genie, Aynalem Yetwale Hiwot, Asiya Muhhamedamin Abagelan, Mulualem Silesh Zerihun.

**Visualization:** Belete Fenta Kebede, Yalemtsehay Dagnaw Genie, Tsegaw Biyazin Tesfa, Aynalem Yetwale Hiwot, Asiya Muhhamedamin Abagelan, Mulualem Silesh Zerihun.

**Writing – original draft:** Belete Fenta Kebede, Yalemtsehay Dagnaw Genie, Tsegaw Biyazin Tesfa, Aynalem Yetwale Hiwot, Asiya Muhhamedamin Abagelan, Mulualem Silesh Zerihun.

**Writing – review & editing:** Belete Fenta Kebede, Yalemtsehay Dagnaw Genie, Tsegaw Biyazin Tesfa, Aynalem Yetwale Hiwot, Asiya Muhhamedamin Abagelan, Mulualem Silesh Zerihun.

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
