## [Decision Letter · Decision Letter 0]

29 Apr 2022

PONE-D-21-31222ADHERENCE TOWARDS COVID-19 PREVENTIVE MEASURES AND ASSOCIATED FACTORS AMONG JIMMA TOWN HIGH SCHOOL STUDENTS, SOUTH-WEST, ETHIOPIA/2021: INSTITUTIONAL-BASED CROSS-SECTIONAL STUDYPLOS ONE

Dear Dr. Kebede,

Thank you for submitting your manuscript to PLOS ONE. After careful consideration, we feel that it has merit but does not fully meet PLOS ONE’s publication criteria as it currently stands. Therefore, we invite you to submit a revised version of the manuscript that addresses the points raised during the review process.

We look forward to receiving your revised manuscript.

Kind regards,

Jianhong Zhou

Staff Editor

PLOS ONE

Journal Requirements:

Whilst you may use any professional scientific editing service of your choice, PLOS has partnered with both American Journal Experts (AJE) and Editage to provide discounted services to PLOS authors. Both organizations have experience helping authors meet PLOS guidelines and can provide language editing, translation, manuscript formatting, and figure formatting to ensure your manuscript meets our submission guidelines. To take advantage of our partnership with AJE, visit the AJE website (http://aje.com/go/plos) for a 15% discount off AJE services. To take advantage of our partnership with Editage, visit the Editage website (www.editage.com) and enter referral code PLOSEDIT for a 15% discount off Editage services.  If the PLOS editorial team finds any language issues in text that either AJE or Editage has edited, the service provider will re-edit the text for free.

Upon resubmission, please provide the following:The name of the colleague or the details of the professional service that edited your manuscriptA copy of your manuscript showing your changes by either highlighting them or using track changes (uploaded as a *supporting information* file)A clean copy of the edited manuscript (uploaded as the new *manuscript* file

Reviewers' comments:

Reviewer's Responses to Questions

**Comments to the Author**

1. Is the manuscript technically sound, and do the data support the conclusions?

Reviewer #1: Yes

2. Has the statistical analysis been performed appropriately and rigorously? 

Reviewer #1: Yes

3. Have the authors made all data underlying the findings in their manuscript fully available?

Reviewer #1: No

4. Is the manuscript presented in an intelligible fashion and written in standard English?

Reviewer #1: No

5. Review Comments to the Author

Reviewer #1: Comment #1: Title

Title: ADHERENCE TOWARDS COVID-19 PREVENTIVE MEASURES AND ASSOCIATED FACTORS AMONG JIMMA TOWN HIGH SCHOOL STUDENTS, SOUTH-WEST, ETHIOPIA/2021: INSTITUTIONAL-BASED CROSS-SECTIONAL STUDY

Better to modify as: ADHERENCE TO COVID-19 PREVENTIVE MEASURES AMONG HIGH SCHOOL STUDENTS IN JIMMA TOWN, SOUTH-WEST ETHIOPIA: INSTITUTIONAL-BASED CROSS-SECTIONAL STUDY

Comment #2: Abstract

Background: The adherence to COVI-19….correct as COVID-19

What is known about the magnitude of COVID-19 in Africa or Ethiopia???

What is known about the adherence to COVID-19 preventive measures in Africa or Ethiopia???

Objective: covid-19…replace with COVID-19 through the manuscript, Ethiopia/2021…use comma

Methods: Institutional based cross-sectional study was conducted on systematically selected high school students attending Jimma town public high schools from July/2021-August/2021…full of repetition of some words or phrases; need revision??

Under methods mention your sample size, and data collection method.

Finally; Significant factors were performed at a significance level of <0.05…use declared than performed.

Result: … About 205(52.8%)… add confidence interval.

Conclusion: In this study the level of adherence to covid-19 preventive measures was low…what is your base to say low?? How 52.8% belongs to be low??

Your recommendation is not in line with your findings???

Comment #3: Introduction

The outbreak was first identified in December 2019 in Wuhan, China…this is not new for readers??

Largest disruption of education systems in history…repeatedly written in 2nd and 3rd paragraph.

Fifth paragraph is too long try to minimize it.

Comment #4: Methods and Materials

Currently, is there preparatory school in Ethiopia??

The duration of the study was From July 2021 to August 2021…mention date??

They were illegible for the study…is it illegible or eligible??

Operational definitions: try to consider three categories (poor, moderate and good) for knowledge, and adherence???

How many items or questions were used to assess knowledge, attitude and adherence???

Why ordinary logistic regression modeling was used???

Comment #5: Results

What is the difference b/n practice and adherence??

In identifying associated to adherence such as factors knowledge and attitude, why you used poor knowledge and unfavorable attitude comparison category than good knowledge and favorable attitude??

6. PLOS authors have the option to publish the peer review history of their article (what does this mean?). If published, this will include your full peer review and any attached files.

Reviewer #1: **Yes: **Demisu Zenbaba

---

## [Author Response · Author response to Decision Letter 0]

17 Jun 2022

We are pleased to hear from you in the shortest possible time ,

thank you

---

## [Decision Letter · Decision Letter 1]

17 Oct 2022

PONE-D-21-31222R1ADHERENCE TO COVID-19 PREVENTIVE MEASURES AMONG HIGH SCHOOL STUDENTS IN JIMMA TOWN, SOUTH-WEST ETHIOPIA: INSTITUTIONAL-BASED CROSS-SECTIONAL STUDYPLOS ONE

Dear Dr. Kebede,

Thank you for submitting your manuscript to PLOS ONE. After careful consideration, we feel that it has merit but does not fully meet PLOS ONE’s publication criteria as it currently stands. Therefore, we invite you to submit a revised version of the manuscript that addresses the points raised during the review process.

We look forward to receiving your revised manuscript.

Kind regards,

Md. Tanvir Hossain

Academic Editor

PLOS ONE

Journal Requirements:

Reviewers' comments:

Reviewer's Responses to Questions

**Comments to the Author**

1. If the authors have adequately addressed your comments raised in a previous round of review and you feel that this manuscript is now acceptable for publication, you may indicate that here to bypass the “Comments to the Author” section, enter your conflict of interest statement in the “Confidential to Editor” section, and submit your "Accept" recommendation.

Reviewer #1: (No Response)

Reviewer #2: (No Response)

2. Is the manuscript technically sound, and do the data support the conclusions?

Reviewer #1: No

Reviewer #2: No

3. Has the statistical analysis been performed appropriately and rigorously? 

Reviewer #1: No

Reviewer #2: I Don't Know

4. Have the authors made all data underlying the findings in their manuscript fully available?

Reviewer #1: Yes

Reviewer #2: Yes

5. Is the manuscript presented in an intelligible fashion and written in standard English?

Reviewer #1: No

Reviewer #2: Yes

6. Review Comments to the Author

Reviewer #1: Comment #2: Abstract

Background:

Indicate what is the gap with adherence to COVID-19 preventive measures among high school students in Africa or Ethiopia???

Please remove number of case of COVID-19 in Africa /Ethiopia from background and you can explain it under introduction.

line number 37: punctuation error

Result: 95% CI: 0.4783- 39 0.5779…write confidence intervals as percent??

AOR: 0.57 (95% CI: 0.37-0.89) write all as (AOR = 0.57, 95% CI: 0.37-0.89)???

Majority of the studies uses good knowledge an positive attitude, why you consider the reverse one??

Conclusion: From where you get national figure, is there systematic review??? How much is it?

what are special strategies??? Is the access to water and soap, only for male students???

Comment #3: Introduction

Try to minimize one page and half.

Comment #4: Methods and Materials

This was my critical comment, but the author fails to revise it??

It is illogical to say 90% and 60% equally as good adherence, for 60% better say moderate??

The appropriate model for these types of category is ordinary logistic regression??

Operational definitions: try to consider three categories (poor, moderate and good) for knowledge, and adherence???

Reviewer #2: This is a timely and relevant study by authors about COVID-19 among students. The findings could be useful at the local and national levels to combat COVID-19. My review is based on the scientific merit of the manuscript and not based on the potential implications of the paper.

Methods:

Authors have mentioned that they used previously used tools to collect the knowledge, attitude, and practice data about COVID-19. The authors didn’t mention the validity and reliability of the tools/questionnaires.

Result:

The authors have found some rather interesting/plausible results. The univariate and adjusted logistic regression models revealed results in different directions.

There is a major flaw in the interpretation of the results. For example in

Table 6 page 15-16; Education status; Grade 9 students had higher odds of adherence to preventive measures compared to grade 10 students. The authors have interpreted it in the other direction. Also, since the 95%CI excludes 1, the p-value must be <0.05. Similarly, the authors have misinterpreted the adjusted odds ratio (most of the no-category have higher odds ratios compared to yes categories. Am I missing anything?

Please pay attention to your results, when you got an odds ratio of 0.57, you said men had lower odds of adherence; which is correct, but when you got an odds ratio of 1.6 for grade 9, shouldn’t you interpret it as l- grade 9 students have 1.6 times higher odds of adherence compared to grade 10? If you do so, you cannot say that higher education is associated with higher odds of adherence. Also, what educational contents regarding COVID/Infectious disease might differ among grade 9 and grade 10 students? Expanding it might help discussion and conclusion.

Also, please interpret your measure of association as an odds ratio. For example, you can mention. The odds were 47%less etc..You have interpreted your odds ratios and risk ratio by directly saying that “…… 2.11 times more likely………..”. You should mention that the ratio is in odds and not directly measured as incidence.

Table 2, Page 13, the institutional factors varied according to the schools rather than the individual participants. You have used variables also in the multivariable analysis (please mention if you assume that students within the same school can have varying values of these characters, if so please justify?) There are other measures to analyze the ecological nature of data.

Minor comments:

The adherence level according to covariates can be best displayed with percentages rather than actual numbers.

Figure 1: It can be explained in a sentence and I don’t think having a figure makes it easier to understand for readers.

Line 258, page 15: The word “multivariate analysis” is not suitable to explain the data analysis in this study. Please use the term adjusted/multiple/or multivariable logistic regression analysis.

Discussion:

I would suggest your reply to my comments about the result section. I think you need to change your discussion significantly after reviewing your results.

The limitation section is poorly written. It is very clear that this is a cross-sectional study and the results could not infer causality. There could be other limitations like the limitation/weakness of the questionnaire if there is any bias due to social desirability. Do you think there is a possibility of recall bias since you asked questions about having COVID to themselves, their family, and their friends?

Also, they developed a cut-off value for knowledge, attitude, and preventive practices based on half score in each category. This can create some limitations to their measure.

Conclusion:

The conclusion is generic and not fully supported by the author's findings. They could be more specific on what implication this study holds to contribute to the battle against COVID-19 in Ethiopia and in their local settings. They found grade 9 students have higher odds of adherence compared to grade 10, why this happened? Given the scenario, how more educational programs could be helpful?

7. PLOS authors have the option to publish the peer review history of their article (what does this mean?). If published, this will include your full peer review and any attached files.

Reviewer #1: **Yes: **Demisu Zenbaba

Reviewer #2: No

---

## [Author Response · Author response to Decision Letter 1]

2 Nov 2022

POINT-BY-POINT RESPONSE TO COMMENTS FROM THE REVIEWERS

Dear: Md. Tanvir Hossain (Academic Editor of the PLOS ONE)

First, we thank you for your response to our submission. In general, we found that the questions and comments from the reviewers were interesting and appropriate. We accept the comments and suggestions. Based on the reviewers comments/ suggestions and we have made corrections accordingly. 

We are pleased to submit an improved version of the manuscript .The changes and modifications have been highlighted in the revised version of the manuscript. A point-by-point response to the comments and questions is provided below. We have used the same letter to write our responses.

ID: PONE-D-21-31222R1

Title: ADHERENCE TO COVID-19 PREVENTIVE MEASURES AMONG HIGH SCHOOL STUDENTS IN JIMMA TOWN, SOUTH-WEST ETHIOPIA: INSTITUTIONAL-BASED CROSS-SECTIONAL STUDY

Sections Reviewers Questions and Suggestions Authors actions and Response

REVIEWER #1; Comment #2 

Abstract Background: 

 Indicate what is the gap with adherence to COVID-19 preventive measures among high school students in Africa or Ethiopia??? Great! We appreciate your comment.

In the abstract of our revised manuscript we had added the statement under here:

This study conducted to fill the information gap on level of adherence to COVID-19 preventive measures among students.

 Please remove number of case of COVID-19 in Africa /Ethiopia from background and you can explain it under introduction. Thank you very much

We agree with your important suggestion, we had removed. 

 line number 37: punctuation error Great! we appreciate you.

It is now corrected/fixed in the revised manuscript 

 Result: 

 1. 95% CI: 0.4783- 39 0.5779…write confidence intervals as percent??

AOR: 0.57 (95% CI: 0.37-0.89) write all as (AOR = 0.57, 95% CI: 0.37-0.89)??? We would like to thank you for your critical issue. 

We accept your indication and we had made corrections on revised manuscript.

 2. Majority of the studies uses good knowledge an positive attitude, why you consider the reverse one?? We appreciate your concern!

Now we had made corrections according to the majority of the studies reporting style and based on your suggestion in our revised manuscript 

 Conclusion: 

 From where you get national figure, is there systematic review??? How much is it? Great! Thank you for your question!

It means from the researchers conducted in other parts of Ethiopia, but currently systematic review and meta-analysis result Published: October 13, 2022. Which shows The pooled estimate of adherence to COVID-19 preventive measures in Ethiopia was 41.15%, which is higher than our study finding.

 What are special strategies??? 

Is the access to water and soap, only for male students??? Thank you 

It is to mean implementing strategies such as increasing, access to health information mainly targeting male students, access to water and soap to school level.

Water and soap access is not only for male students, it is for all school students.

Comment #3 Introduction 

 1. Try to minimize one page and half. Thank you for your appropriate suggestion!

We had made corrections 

Comment #4 Comment #4: Methods and Materials 

 This was my critical comment, but the author fails to revise it??

It is illogical to say 90% and 60% equally as good adherence, for 60% better say moderate?? Thank you!

Dear reviewer! Thank you for your important indication how it could be corrected. We authors agree and we had made all corrections based on your indication in the revised manuscript.

 2. The appropriate model for these types of category is ordinary logistic regression?? Thank you for your suggestion 

We had made analysis again using ordinary logistic regression after your second comment, now all associations made using ordinary logistic regression model again. 

 Operational definitions: 

 1. Try to consider three categories (poor, moderate and good) for knowledge, and adherence??? Thank you for your important suggestion

Therefore, we authors agree to change the category of variables according to your suggestion, and we had made the regression based on the three category. The modifications had experienced on the results.

Thank you again for understanding us.

REVIEWER #2; Comment #1 Authors actions and Response for each specific issues 

This is a timely and relevant study by authors about COVID-19 among students. 

The findings could be useful at the local and national levels to combat COVID-19. 

My review is based on the scientific merit of the manuscript and not based on the potential implications of the paper Thank you very much 

Comment Methods: 

 Authors have mentioned that they used previously used tools to collect the knowledge, attitude, and practice data about COVID-19. The authors did not mention the validity and reliability of the tools/questionnaires. Thank you for raising important point that was missed,

we had made additional explanations, as follows 

Data collection instrument and quality control 

After reviewing different studies done before (1,2,14–16) the questionnaire was adapted to address the objectives of the study. To assure the quality of the data, the tool was prepared first in English and then translated into the local language (Amharic and Afan Oromo) by language experts in English and both local languages. The reliability and validity of the instrument was checked using Cronbach alpha (0.751) and subject expertise (public health $ nursing) respectively. A pre-test was done on 5% of the total sample size in another school, which was not selected for actual data collection. Then modifications such as wording, rephrasing, adding and deleting of some information for clarity were made on the tool accordingly. Data collectors and supervisors were trained on the data collection process for one day. The data were checked for completeness and consistency of information by the principal investigator. 

 Result: 

 The authors have found some rather interesting/plausible results. 

The univariate and adjusted logistic regression models revealed results in different directions. We appreciate your insight full comments!

 Dear reviewer, apologize! We had made check-up to find how we made mistakes during analysis, and now we corrected and fix it in the revised manuscript.

Thank you again 

 There is a major flaw in the interpretation of the results.

 For example in Table 6 page 15-16; Education status; Grade 9 students had higher odds of adherence to preventive measures compared to grade 10 students. 

The authors have interpreted it in the other direction. Thank you dear reviewer for your critical/insightful observation

We authors note your point and we agree you are right. 

we had made all the analysis again using three categories to the outcome variable according to suggestions from other(1st) reviewer, we feel that now your issue has been solved in the revised manuscript,

Thank you again! 

 Also, since the 95%CI excludes 1, the p-value must be <0.05. 

Similarly, the authors have misinterpreted the adjusted odds ratio (most of the no-category have higher odds ratios compared to yes categories. Am I missing anything? Great, we appreciate all concerns and we share it.

We had made all the analysis again and now it is corrected based on the recommendations from both reviewers in the revised manuscript. Table 7.

 Please pay attention to your results, when you got an odds ratio of 0.57, you said men had lower odds of adherence; which is correct, but when you got an odds ratio of 1.6 for grade 9, shouldn’t you interpret it as l- grade 9 students have 1.6 times higher odds of adherence compared to grade 10? 

If you do so, you cannot say that higher education is associated with higher odds of adherence.

 Also, what educational contents regarding COVID/Infectious disease might differ among grade 9 and grade 10 students? Expanding it might help discussion and conclusion. We thank you!

We try to fix all the issues that you indicate in the revised manuscript after running the statistical analysis again. 

We re-write all the results and discussions based on the new statistical analysis result.

From the new statistical analysis report educational status is not associated with level of adherence, so all things are solved. 

 Also, please interpret your measure of association as an odds ratio. For example, you can mention. 

The odds were 47%less etc..You have interpreted your odds ratios and risk ratio by directly saying that “…… 2.11 times more likely………..”. You should mention that the ratio is in odds and not directly measured as incidence. It is now fixed based on your recommendation in the revised manuscript 

Thank you again!

 Table 2, Page 13, the institutional factors varied according to the schools rather than the individual participants. You have used variables also in the multivariable analysis (please mention if you assume that students within the same school can have varying values of these characters, if so please justify?) There are other measures to analyze the ecological nature of data Thank you for raising important point!

 We authors did not assume that students within the same school could have varying values of these characters; unfortunately, the study was conducted at different schools, so students from different schools may have different institutional factors. In addition, we have used variables in the multivariable analysis after reviewing of many other studies done previously.

Here we are ready to correct if we did mistakes again and ready to hear from you.

Thank you again!

Minor comments 

 The adherence level according to covariates can be best displayed with percentages rather than actual numbers. Thank you 

accepted and corrected in the revised manuscript 

 Figure 1: It can be explained in a sentence and I don’t think having a figure makes it easier to understand for readers. Great 

It has been corrected and submitted separately

 Line 258, page 15: The word “multivariate analysis” is not suitable to explain the data analysis in this study. Please use the term adjusted/multiple/or multivariable logistic regression analysis. Thank you very much 

now it is corrected 

 Discussion: 

 I would suggest your reply to my comments about the result section. I think you need to change your discussion significantly after reviewing your results. Thank you again for suggestion

we have made changes according to the results changed accordingly

 The limitation section is poorly written. It is very clear that this is a cross-sectional study and the results could not infer causality. There could be other limitations like the limitation/weakness of the questionnaire if there is any bias due to social desirability. Do you think there is a possibility of recall bias since you asked questions about having COVID to themselves, their family, and their friends?

Also, they developed a cut-off value for knowledge, attitude, and preventive practices based on half score in each category. This can create some limitations to their measure we would like to appreciate all your points raised

We had made some modifications according to your suggestion. 

now the limitation section rewritten as :

The following limitation can be considered in relation to this study. It is very clear that this is a cross-sectional study and the results could not infer causality. Social-desirability bias may one limitation that students might respond to the socially acceptable responses despite their actual practice is poor. There is also a possibility of recall bias since we asked questions about having COVID-19 to themselves, their family, and their friends. In addition, it has been developed a cut-off value for attitude of respondents based on half score in each category, this can create some limitations to their measure. 

 Conclusion 

 The conclusion is generic and not fully supported by the author's findings. They could be more specific on what implication this study holds to contribute to the battle against COVID-19 in Ethiopia and in their local settings. They found grade 9 students have higher odds of adherence compared to grade 10, why this happened. Given the scenario, how more educational programs could be helpful? Thank you for your important points.

We re-written the entire conclusion again and modified in the revised manuscript according to your suggestion.

Summary: we authors feel that the reviewer, questions, comments and suggestions were very important and have a great contribution to the improvement of the paper. We have made corrections and we try to fix the typographical and language edition. This corrections and changes highlighted with green colour . 

Thank you again!

---

## [Decision Letter · Decision Letter 2]

1 Dec 2022

ADHERENCE TO COVID-19 PREVENTIVE MEASURES AMONG HIGH SCHOOL STUDENTS IN JIMMA TOWN, SOUTH-WEST ETHIOPIA: INSTITUTIONAL-BASED CROSS-SECTIONAL STUDY

PONE-D-21-31222R2

Dear Dr. Kebede,

We’re pleased to inform you that your manuscript has been judged scientifically suitable for publication and will be formally accepted for publication once it meets all outstanding technical requirements.

Kind regards,

Md. Tanvir Hossain

Academic Editor

PLOS ONE

Reviewers' comments:

Reviewer's Responses to Questions

**Comments to the Author**

1. If the authors have adequately addressed your comments raised in a previous round of review and you feel that this manuscript is now acceptable for publication, you may indicate that here to bypass the “Comments to the Author” section, enter your conflict of interest statement in the “Confidential to Editor” section, and submit your "Accept" recommendation.

Reviewer #2: All comments have been addressed

2. Is the manuscript technically sound, and do the data support the conclusions?

Reviewer #2: Yes

3. Has the statistical analysis been performed appropriately and rigorously? 

Reviewer #2: Yes

4. Have the authors made all data underlying the findings in their manuscript fully available?

Reviewer #2: Yes

5. Is the manuscript presented in an intelligible fashion and written in standard English?

Reviewer #2: Yes

6. Review Comments to the Author

Reviewer #2: The revised manuscript stands in good condition. I am happy to recommend for the acceptance of this paper. However, i would like to suggest for very few minor revisions.

Statistical analysis: I suppose when you mentioned ordinary logistic regression you meant "Ordinal". Please correct the terminology as the ordinary regression means another analysis method.

First line of the study area and period includes a typo please correct High Schools to high schools.

In the abstract result section: please elaborate the term "sex" you can say female sex or female gender while explaining the higher odds.

1. Table 7: I was wondering why did not you adjust for presence of hand washing facility(p=0.095) in the multivariable regression? You said the cut-off for adjustment was p<0.25.

7. PLOS authors have the option to publish the peer review history of their article (what does this mean?). If published, this will include your full peer review and any attached files.

Reviewer #2: No

---

## [Editor Report · Acceptance letter]

4 Dec 2022

PONE-D-21-31222R2 

ADHERENCE TO COVID-19 PREVENTIVE MEASURES AMONG HIGH SCHOOL STUDENTS IN JIMMA TOWN, SOUTH-WEST ETHIOPIA: INSTITUTIONAL-BASED CROSS-SECTIONAL STUDY 

Dear Dr. Kebede:

I'm pleased to inform you that your manuscript has been deemed suitable for publication in PLOS ONE. Congratulations! Your manuscript is now with our production department. 

Kind regards, 

on behalf of

Dr. Md. Tanvir Hossain 

Academic Editor

PLOS ONE